# Chiral Loop Quantum Supergravity and Black Hole Entropy

**Konstantin Eder** [†] [iD] **and Hanno Sahlmann** *,[†] [iD]

Institute for Quantum Gravity (IQG), Department of Physics, Friedrich-Alexander-Universität Erlangen-Nürnberg (FAU), 91054 Erlangen, Germany; konstantin.eder@fau.de
* Correspondence: hanno.sahlmann@fau.de
† These authors contributed equally to this work.

**Abstract:** Recent work has shown that local supersymmetry on a spacetime boundary in $\mathcal{N}$-extended AdS supergravity in chiral variables implies coupling to a boundary $\mathrm{OSp}(\mathcal{N}|2)_{\mathbb{C}}$ super Chern–Simons theory. Consequently there has been a proposal to define and calculate the entropy $S$ for the boundary, in the supersymmetric version of loop quantum gravity, for the minimal case $\mathcal{N} = 1$, via this super Chern–Simons theory. We give an overview of how supergravity can be treated in loop quantum gravity. We review the calculation of the dimensions of the quantum state spaces of $\mathrm{UOSp}(1|2)$ super Chern–Simons theory with punctures, and its analytical continuation, for the fixed quantum super area of the surface, to $\mathrm{OSp}(1|2)_{\mathbb{C}}$. The result is $S = a_H/4$ for large (super) areas. Lower order corrections can also be determined. We begin also a discussion of the statistical mechanics of the surface degrees of freedom by calculating the grand canonical partition function at zero chemical potential. This is a new result.

**Keywords:** chiral supergravity; black hole entropy; loop quantum gravity

## 1. Introduction

Since there are indications that horizons can meaningfully be assigned a thermodynamic entropy [1–3], the challenge is to explain it as the von Neumann-type entropy of a quantum description of the black hole. This challenge has been met with some measure of success in string theory (the entropy of BPS black holes [4,5] for example) and loop quantum gravity (entropy of isolated horizons, for example [6–13]). The two approaches are very different in nature and are not applicable to the same type of black hole. In this article, we review and expand on recent work [14] that starts to bridge this gap. The theory considered is $\mathcal{N} = 1, D = 4$ supergravity, on a manifold $M$ with a boundary $H = \partial M$ with special boundary conditions. The boundary degrees of freedom are described by a super Chern–Simons theory. We define the entropy of the boundary as the logarithm of the dimension of the Chern–Simons state space. The level of the super Chern–Simons theory is imaginary, and the structure group is non-compact, so the properties of the state space are not known. We use an analytic continuation from a compact real form of the structure group, following [15–17]. The Chern–Simons theory is constrained by the super area $a_H$ of the boundary, and hence the entropy becomes area-dependent. The result is

$$S = \frac{a_H}{4l_p^2} + \mathcal{O}(\sqrt{a_H}/l_p), \tag{1}$$

where $a_H$ is the diffeomorphism and gauge invariant measure of area[1] in the supergeometric setting.

We are using variables that were first proposed in [18] and whose geometric meaning was recently clarified in [19–21]. A super Chern–Simons theory as a source of entropy was first considered in [6]. Supergravity with canonical loop quantum gravity methods has also been considered in [22], and using different variables in [23–25]. The spinfoam

approach for $D = 3$ was developed in [26–28]. Our approach follows [6,18,22,29] in keeping supersymmetry manifest, but it goes further. We use the geometric understanding for the super Ashtekar connection developed in [19–21]. The boundary conditions follow from demanding supersymmetry at the boundary [21]. Our treatment is the first one that uses state counting in super Chern–Simons theory, as far as we know. Additionally, this is the first time that the Bekenstein–Hawking area law is verified within the supersymmetric setting.

We work in the Hamiltonian formulation of supergravity. The canonical variables are a supersymmetric generalization [18–21] of the chiral Ashtekar connection $\mathcal{A}^+$, and the corresponding canonical momentum, the super electric field $\mathcal{E}$. As the classical Ashtekar variables can be obtained from the Palatini action augmented with the Holst term, the variables used here can be derived from a Holst-type modification of the MacDowell–Mansouri action [19,21] for supergravity. The structure group in this formulation is OSp(1|2). We consider this theory in the presence of a causal boundary $H = \partial M$ of spacetime manifold $M$ playing the role of the horizon. In the canonical formulation, the fields take values on a spatial slice $\Sigma$ of $M$, with boundary $\Delta = \Sigma \cap H$. The requirement of local supersymmetry also on the boundary *uniquely* fixes a supersymmetric boundary term that is given by an OSp(1|2) Chern–Simons theory, and boundary conditions

$$F(\underleftarrow{\mathcal{A}^+}) \propto \underleftarrow{\mathcal{E}}, \tag{2}$$

linking the curvature $F$ of the chiral super connection $\mathcal{A}^+$ and the super-electric field $\mathcal{E}$ on the boundary $\Delta$.

The arrows under the symbols denote pullback to the boundary. As in the non-supersymmetric case [8], the idea is to quantize bulk and boundary separately and couple them via the boundary condition. The line-like excitations of the gravitational field in the bulk can end on the boundary $\Delta$. There, they couple to the Chern–Simons theory by creating particle-like curvature defects.

The quantum theory in the bulk is built in analogy to that of the non-supersymmetric theory. There are unresolved issues, such as the construction of the full Hilbert space from a projective limit [19], but only states on fixed graphs are relevant for the entropy calculation; hence this does not present an obstacle for our purposes. On the other hand, many aspects of the non-supersymmetric theory suitably generalize to the supersymmetric situation.[2] In particular, one can obtain a graded holonomy-flux algebra, a supersymmetric generalizations of spin networks, and a supersymmetric area operator [14], see also [22]. OSp(1|2) is non-compact, leading to various technical problems. We, therefore, start from the Chern–Simons theory of a compact real form of this group, UOSp(1|2) and use analytical continuation in the corresponding Verlinde-type formula that is counting its states, following and generalizing [15,16].

Let us explain in what respects the present account of black hole entropy differs from the one in string theory. Our setup involves a large class of surfaces that carry local supersymmetry, whereas, in string theory, the relevant surfaces are horizons of BPS black holes; hence there is greater geometric restriction but also specificity. Additionally, in the present calculation, we start from the quantum states of a super Chern–Simons theory and couple to a quantum theory that is invariant under local supersymmetry. Hence fermionic degrees of freedom are part of the calculation from the beginning, whereas they do not seem to play a direct role in the string theory calculation. One could hope that these fermionic fields have interesting consequences in a more detailed treatment, see for example [30], where it was demonstrated that a nontrivial supercharge can lead to an additional term in the first law of black hole mechanics.

## 2. Chiral Supergravity from a Modified MacDowell–Mansouri Action

We use the Cartan geometric description of pure AdS Holst-supergravity with $\mathcal{N}$-extended supersymmetry with $\mathcal{N} = 1, 2$. Details are in [19–21] (see also [31,32] using standard variables, and [26–28] in the context of spinfoam models in $D = 3$).

Pure AdS (Holst-)supergravity can be described in terms of a super Cartan geometry modeled on the super Klein geometry $(\mathrm{OSp}(\mathcal{N}|4), \mathrm{Spin}^+(1,3) \times \mathrm{SO}(\mathcal{N}))$ with super Cartan connection

$$\mathcal{A} = e^I P_I + \frac{1}{2}\omega^{IJ} M_{IJ} + \frac{1}{2}\hat{A}_{rs} T^{rs} + \Psi_r^\alpha Q_\alpha^r. \tag{3}$$

Here we have split the connection in terms of generators adapted to subalgebras of $\mathfrak{osp}(\mathcal{N}|4)$. The $P$ generate translations, $M$ rotations and boosts, $T$ generate the R-symmetry. These are all even generators. The odd generators are the $Q$. Along with this decomposition of $\mathfrak{osp}(\mathcal{N}|4)$ goes the decomposition of $\mathcal{A}$ into the tetrad $e$, the $\mathfrak{so}(3,1)$ connection $\omega$, $\mathrm{SO}(\mathcal{N})$ gauge field $\hat{A}$, and fermion fields[3] $\Psi$. This connection can be used in order to formulate a Yang–Mills-type action principle for Holst-supergravity. One introduces a $\beta$-deformed inner product $\langle \cdot \wedge \cdot \rangle_\beta$ on $\mathfrak{g} \equiv \mathfrak{osp}(\mathcal{N}|4)$-valued differential forms on the underlying spacetime manifold $M$ with $\beta$ the Barbero-Immirzi parameter obtained via contraction of the Ad-invariant supertrace with a $\beta$-dependent operator $\mathbf{P}_\beta$ on $\Omega^2(M, \mathfrak{g})$ (the precise form of this operator does not matter in what follows; for more details see [19,21]). Using this inner product, the Holst–MacDowell–Mansouri action of $\mathcal{N}$-extended pure Ads Holst-supergravity takes the form

$$S_{\mathrm{H\text{-}MM}}^\beta(\mathcal{A}) = \frac{L^2}{\kappa} \int_M \langle F(\mathcal{A}) \wedge F(\mathcal{A}) \rangle_\beta, \tag{4}$$

with $F(\mathcal{A})$ the Cartan curvature of $\mathcal{A}$.

In the chiral limit of the theory corresponding to an imaginary $\beta = -i$, the action (4) becomes manifestly invariant under an enlarged $\mathrm{Osp}(\mathcal{N}|2)_{\mathbb{C}}$-gauge symmetry: The operator $\mathbf{P}_{-i}$ decomposes as $\mathbf{P}_{-i} = \tilde{\mathbf{P}}_{-i} \circ \mathbf{P}^{\mathfrak{osp}(\mathcal{N}|2)}$ with $\mathbf{P}^{\mathfrak{osp}(\mathcal{N}|2)} : \mathfrak{osp}(\mathcal{N}|4) \to \mathfrak{osp}(\mathcal{N}|2)_{\mathbb{C}}$ the projection operator onto the (complexified) chiral sub superalgebra $\mathfrak{osp}(\mathcal{N}|2)_{\mathbb{C}}$ of $\mathfrak{g}$. Applying this projection operator on the super Cartan connection (3) yields the super Asthekar connection

$$\mathcal{A}^+ := \mathbf{P}^{\mathfrak{osp}(\mathcal{N}|2)} \mathcal{A} = A^{+i} T_i^+ + \frac{1}{2}\hat{A}_{rs} T^{rs} + \psi_r^A Q_A^r. \tag{5}$$

In the chiral limit, the action becomes

$$S_{\mathrm{H\text{-}MM}}^{\beta=-i}(\mathcal{A}) = \frac{i}{\kappa} \int_M \left( \langle F(\mathcal{A}^+) \wedge \mathcal{E} \rangle + \frac{1}{4L^2} \langle \mathcal{E} \wedge \mathcal{E} \rangle \right) + S_{\mathrm{bdy}}, \tag{6}$$

with $\mathcal{E}$ the super electric field canonically conjugate to the super Asthekar connection $\mathcal{A}^+$ and transforming under the Adjoint representation of $\mathrm{OSp}(\mathcal{N}|2)_{\mathbb{C}}$. $L$ is related to the cosmological constant, $L^2 = -3/\Lambda_{\mathrm{cos}}$. The boundary action is given by

$$S_{\mathrm{bdy}}(\mathcal{A}^+) = \frac{k}{4\pi} \int_H \langle \mathcal{A}^+ \wedge \mathrm{d}\mathcal{A}^+ + \frac{1}{3}\mathcal{A}^+ \wedge [\mathcal{A}^+ \wedge \mathcal{A}^+] \rangle, \tag{7}$$

i.e., the action of a $\mathrm{OSp}(\mathcal{N}|2)_{\mathbb{C}}$ super Chern–Simons theory with (complex) Chern–Simons level $k = i4\pi L^2/\kappa = -i12\pi/\kappa\Lambda_{\mathrm{cos}}$. As discussed in detail in [19,21], this boundary action arising from (4) in the chiral limit is unique if one imposes supersymmetry invariance at the boundary (see also [31,32]).

The decomposition of (6) into a bulk and boundary action leads to an additional boundary condition coupling bulk and boundary degrees of freedom in order to ensure consistency with the equations of motion of the full theory,

$$\underleftarrow{F(\mathcal{A}^+)} = -\frac{1}{2L^2} \underleftarrow{\mathcal{E}}. \tag{8}$$

This is an equation for two super two-forms on the boundary, obtained by pull-back from the bulk. We note the close analogy to the boundary conditions in the non-

supersymmetric theory [12], which also connects curvature and the (two-form dual of) the densitized triad field on the boundary. An important difference is the constant, which in our case, is related to the cosmological constant.

## 3. Quantum Theory of the Bulk

We will now turn to the quantum theory of the bulk. Many structures found in the non-supersymmetric theory generalize to the present setting. For details see [19].

### 3.1. Super Spin Networks and the Super Area Operator

In standard LQG, for the construction of the bulk Hilbert space $\mathfrak{H}_\gamma^{\text{bulk}}$ associated to a graph $\gamma$ embedded in the spatial slices $\Sigma$ of the spacetime manifold $M = \mathbb{R} \times \Sigma$, one considers spin network states, a class of states invariant under local gauge transformations. They are constructed via the contraction of matrix coefficients of irreducible representations of the underlying gauge group. In the case where the bosonic group is compact, these states form an orthonormal basis of the full Hilbert space. This follows from the well-known Peter–Weyl theorem, which is valid for compact bosonic groups.

To have a well-defined action of holonomy operators, it is necessary that the set of representations on which the quantum theory is built from a tensor category under the tensor product. In this way, matrix elements of holonomies in representations in this category map representations within this category. Finite-dimensional irreducible representations of the orthosymplectic series $\text{OSp}(\mathcal{N}|2)$ for $\mathcal{N} = 1, 2$ have been investigated (e.g., [33–36], see also [26–28]). For the case $\mathcal{N} = 1$, these representations form a subcategory closed under the tensor product. The corresponding spin network states have been studied for instance in [22,28,29]. For the case $\mathcal{N} = 2$, the subclass of typical representations form such a category (see [27,33]).

We now describe the construction of the super spin network states for a suitable subclass $\mathcal{P}_{\text{adm}}$ of irreducible representations. They can be finite- or infinite-dimensional, and they include those constructed in [14] of $\text{OSp}(\mathcal{N}|2)$ with $\mathcal{N} = 1, 2$. For any subset $\vec{\pi} := \{\pi_e\}_{e \in E(\gamma)} \subset \mathcal{P}_{\text{adm}}$, we define the cylindrical function $T_{\gamma, \vec{\pi}, \vec{m}, \vec{n}}$ via

$$T_{\gamma, \vec{\pi}, \vec{m}, \vec{n}}[\mathcal{A}^+] := \prod_{e \in E(\gamma)} \pi_e(h_e[\mathcal{A}^+])^{m_e}{}_{n_e}, \tag{9}$$

where, for any edge $e \in E(\gamma)$, $h_e[\mathcal{A}^+]$ denotes the super holonomy (parallel transport) of the connection $\mathcal{A}^+$ along $e$ (see [19,37]) and $(\pi_e)^{m_e}{}_{n_e}$ denote certain matrix coefficients of the representation $\pi_e \in \mathcal{P}_{\text{adm}}$. In order to obtain a gauge invariant state, at each vertex $v \in V(\gamma)$ of the graph $\gamma$, we have to contract (9) with an intertwiner $I_v$ projecting onto the trivial representation at any vertex. As a result, the so-constructed state transforms trivially under local gauge transformations and thus indeed forms a gauge-invariant state which we call a (gauge-invariant) super spin network state. We take these states as a basis of the state space of the bulk theory. We assume that an inner product can be found that turns this space into a super Hilbert space $\mathfrak{H}_\gamma^{\text{bulk}}$.

In the space of super spin networks, one can introduce a gauge-invariant operator in analogy to the area operator in ordinary LQG. More precisely, since the super electric field $\mathcal{E}$ defines a $\mathfrak{g}$-valued 2-form, for any oriented (semianalytic) surface $S$ embedded in $\Sigma$, one can define the *graded* or *super area* $\text{gAr}(S)$ via

$$\text{gAr}(S) := \sqrt{2} \int_S \sqrt{\langle \mathcal{E}_S, \mathcal{E}_S \rangle}, \tag{10}$$

where, in analogy to [38–40], $\mathcal{E}_S$ is defined as the unique $\mathfrak{g}$-valued function on $S$ such that $\iota_S^* \mathcal{E} = \mathcal{E}_S \text{vol}_S$. For the special case $\mathcal{N} = 1$, the expression (10) coincides with the super area as considered in [22]. Here, the prefactor $\sqrt{2}$ has been chosen such that in the case of vanishing fermionic degrees of freedom, the super area reduces to the standard area of $S$ in ordinary Riemannian geometry.

By definition, the quantity (10) solely depends on the super electric field, which defines a phase space variable. Thus, we can implement it in the quantum theory (see [14] for more details). As a result, for $\mathcal{N} = 1$, it follows for instance in the case that the surface $S$ intersects the graph $\gamma$ of a (gauge-invariant) super spin network state $T_{\gamma, \vec{\pi}, \vec{m}, \vec{n}}$ labeled by super spin quantum numbers $j \in \mathbb{C}$ corresponding to the principal series representations of $\mathrm{OSp}(1|2)$ as constructed in [14] in a single divalent vertex $v \in V(\gamma)$ that the action of super area operator is given by

$$\widehat{\mathrm{gAr}}(S) T_{\gamma, \vec{\pi}, \vec{m}, \vec{n}} = -8\pi i l_p^2 \sqrt{j\left(j + \frac{1}{2}\right)} T_{\gamma, \vec{\pi}, \vec{m}, \vec{n}}, \tag{11}$$

with $j \in \mathbb{C}$ the superspin quantum number labeling the edge $e \in E(\gamma)$ intersecting the surface $S$. For $j \in \frac{\mathbb{N}_0}{2}$, this coincides with the result of [22].

According to (11), the super area operator has complex eigenvalues. This is to be expected since we are working with a complexification of supergravity. However, we are interested in the real sector, corresponding to real supergravity. Reality conditions require a real spatial metric in particular. Importantly, there are principal series representations that have purely imaginary eigenvalues of the Casimir and hence real super area. This is the case for representations labeled by superspin quantum numbers of the form

$$j = -\frac{1}{4} + is \text{ with } s \in \mathbb{R}. \tag{12}$$

In those representations,

$$\widehat{\mathrm{gAr}}(S) T_{\gamma, \vec{\pi}, \vec{m}, \vec{n}} = 8\pi l_p^2 \sqrt{s^2 + \frac{1}{16}} \, T_{\gamma, \vec{\pi}, \vec{m}, \vec{n}}. \tag{13}$$

This is analogous to the non-supersymmetric theory for complex Ashtekar variables, where one can also find representations with real area eigenvalues [15].

### 3.2. Coupling of Boundary and Bulk

As described in Section 3.1 the quantum excitations of the bulk degrees of freedom are represented by super spin network states associated with the gauge supergroup $\mathrm{OSp}(\mathcal{N}|2)_{\mathbb{C}}$. On the other hand, in Section 2, we have explained that the boundary theory is described in terms of a $\mathrm{OSp}(\mathcal{N}|2)_{\mathbb{C}}$ super Chern–Simons theory. Hence, for a given finite graph $\gamma$ embedded in $\Sigma$, we define the Hilbert space $\mathfrak{H}_{\mathrm{full}, \gamma}$ w.r.t. $\gamma$ of the full theory as the tensor product

$$\mathfrak{H}_{\gamma}^{\mathrm{full}} = \mathfrak{H}_{\gamma}^{\mathrm{bulk}} \otimes \mathfrak{H}_{\gamma}^{\mathrm{bdy}}, \tag{14}$$

with $\mathfrak{H}_{\gamma}^{\mathrm{bulk}}$ the Hilbert space of the quantized bulk degrees of freedom as constructed in Section 3.1 and $\mathfrak{H}_{\gamma}^{\mathrm{bdy}}$ the Hilbert space corresponding to the quantized super Chern–Simons theory on the boundary.

On this Hilbert space, we have to implement the boundary condition (8). Let $p \in \mathcal{P}_{\gamma} := \gamma \cap \Delta^4$ denote a puncture and let $D_{\epsilon}(p)$ be a disk in $\Delta$ around $p$ with radius $\epsilon$ in some fiducial metric. We define

$$\mathcal{E}[\alpha](p) := \lim_{\epsilon \to 0} \int_{D_{\epsilon}(p)} \langle \alpha, \mathcal{E} \rangle, \quad F[\alpha](p) := \lim_{\epsilon \to 0} \int_{D_{\epsilon}(p)} \langle \alpha, F(\mathcal{A}^+) \rangle, \tag{15}$$

where $\alpha$ is an arbitrary tuple of smearing functions. These quantities (or, in the case of the curvature, a suitable function thereof) can be promoted to well-defined operators in the quantum theory. Thus, (8) yields the additional constraint equation

$$\mathbb{1} \otimes \widehat{F}_{\underline{A}}(p) = -\frac{2\pi i}{\kappa k} \widehat{\mathcal{E}}_{\underline{A}}(p) \otimes \mathbb{1} \tag{16}$$

at each puncture $p \in \mathcal{P}_\gamma$, in analogy to the bosonic theory [9,12]. The quantized super electric flux $\widehat{\mathcal{E}}_{\underline{A}}(p)$ acts as a derivation on the cylindrical functions of the connection $\mathcal{A}^+$. As such, the flux is carried by the spin network edges, and the flux into the boundary is concentrated at the punctures. Under the boundary conditions (16), this means that the Chern–Simons curvature is similarly concentrated at the punctures. This is perfectly compatible with the Chern–Simons equations of motion in the presence of curvature defects at the punctures $\mathcal{P}_\gamma$. The excitations of the Chern–Simons theory are thus described by a Hilbert space given as a conformal block of a super conformal QFT. The dimension of these conformal blocks is central to the computation of the black hole entropy in the non-supersymmetric theory [9,12].

## 4. Entropy Calculation

A natural measure for the entropy of the boundary is (the logarithm of) the size of the state space of the boundary theory, in our case $OSp(1|2)_{\mathbb{C}}$ Chern–Simons theory with punctures. The level of the Chern–Simons theory is complex. Unfortunately, little is known about Chern–Simons theory with non-compact structure groups. It is also not clear how to treat the purely imaginary level. However, these problems also arise in the chiral theory in the non-supersymmetric case [16] (and also in string theory, see [41]). The approach we chose is to adapt the method of [16]. We consider the Chern–Simons theory with the compact real form $UOSp(1|2)$ of $OSp(1|2)_{\mathbb{C}}$ as structure group and real level, and suitably analytically continue to $OSp(1|2)_{\mathbb{C}}$.

### 4.1. Super Characters of $UOSp(1|2)$ and the Verlinde Formula

Let us consider the Chern–Simons theory with compact gauge supergroup given by the unitary orthosymplectic group $UOSp(1|2) = U(1|2) \cap OSp(1|2)$ and integer Chern–Simons level $k = -12\pi/\kappa\Lambda_{\text{cos}}$ and punctures labeled by finite-dimensional irreducible representations $\vec{j}$ of $UOSp(1|2)$ with $j \in \frac{\mathbb{N}_0}{2}$. We compute the number $\mathcal{N}_k(\vec{j})$ of Chern–Simons degrees of freedom given by the dimension of the superconformal blocks. We then perform an analytic continuation by replacing $j \to j = -\frac{1}{4} + is$ for some $s \in \mathbb{R}$ for each $j \in \vec{j}$ as well as $k \to ik$ in $\mathcal{N}_k(\vec{j})$. To simplify the discussion, we assume that the boundary $H$ is topologically of the form $\mathbb{R} \times \mathbb{S}^2$, that is, the 2-dimensional slices $\Delta_t$ are topologically equivalent to 2-spheres. We also take the limit $k \to \infty$ corresponding to a small cosmological constant $\Lambda_{\text{cos}}$. In this limit, the dimension of the state space $\mathcal{N}_\infty(\vec{j})$ is given by the number of $UOSp(1|2)$ gauge-invariant states, which in turn are described by invariant tensors in the tensor product of the representations incoming to the boundary. These tensors span the space of trivial sub-representations contained in the tensor product representation $\bigotimes_j \pi_j$. Let $n$ be the number of punctures. By subdividing $\vec{j}$ into $p \le n$ subfamilies $(n_l, j_l)$, $l = 1, \ldots, p$, consisting of $0 < n_l \le n$ punctures labeled by $j_l \in \vec{j}$, and using the orthogonality of characters, one can express the supertrace of the projector onto the invariant subspace in the tensor product as (for details see [14])

$$\mathcal{N}_\infty(\{n_l, j_l\}) = \frac{1}{2\pi} \int_0^\pi \mathrm{d}\theta \, \sin^2(2\theta) \prod_{l=1}^p \left( \frac{\cos(d_{j_l}\theta)}{\cos\theta} \right)^{n_l} \left[ 4 - n + \sum_{i=1}^p n_i d_{j_i} \frac{\tan(d_{j_i}\theta)}{\tan\theta} \right], \tag{17}$$

with $d_j := 4j + 1$ the dimension of the spin-$j$ representation of $UOSp(1|2)$.

### 4.2. The Monochromatic Case

We use (17) to compute the entropy associated to the boundary by performing an analytic continuation $OSp(1|2)_{\mathbb{C}}$ of chiral LQSG: We replace the superspin quantum numbers $j \in \vec{j}$ in (17) by $j \to -\frac{1}{4} + is$, i.e., quantum numbers corresponding to the principal series with respect to which the super area operator has purely real eigenvalues. We will first

work with a simplifying assumption. We will stipulate that the punctures are all labeled by the same representation, i.e., the same quantum number $j$. This case will be referred to as the *monochromatic case* in the following. Then, by replacing $j \to -\frac{1}{4} + is$ for some $s \in \mathbb{R}_{>0}$ in (17) for the special case $p = 1$ and using $d_j = i4s =: i\tilde{s}$ as well as $\cos(ix) = \cosh(x)$ and $\sin(ix) = i\sinh(x)$, we can write the analytically continued version of (17) as a contour integral in the complex plane (for details see [14])

$$\mathcal{I}_\infty = \frac{1}{2\pi} \int_{\mathcal{C}} dz\, \mu(z) \left(4 - n\left[1 + \tilde{s}\frac{\tan(\tilde{s}z)}{\tanh z}\right]\right) \exp\left(n \ln\left(\frac{\cos(\tilde{s}z)}{\cosh z}\right)\right), \quad (18)$$

with density $\mu(z) := i\sinh^2(2z)$. $\mathcal{C}$ is a path from 0 to $i\pi$. This integral can be approximately calculated in the limit of macroscopic black holes. To this end, note that macroscopic black holes are expected to have a large number $n$ of punctures. This is borne out both in models for black hole horizons from loop quantum gravity (see for example [10,15]), and self-consistently in the present case (see (51)). In the limit $n \to \infty$, we can apply the method of steepest descent, which requires us to deform $\mathcal{C}$ to pass through all critical points $z_c$, in the direction of steepest descent. From (18), one infers that the only part in the state sum that grows exponentially in the macroscopic limit is given by the last factor in the integrand. Hence, in this limit, the relevant action is given by

$$\mathcal{S}(z) = \ln\left(\frac{\cos(\tilde{s}z)}{\cosh z}\right). \quad (19)$$

Its critical points are all located in the interval $[0, i\pi]$ on the imaginary axis. Evaluating $\mathcal{S}'(z_c) = 0$ gives

$$\tilde{s}\tan(\tilde{s}z_c) = -\tanh(z_c) \quad (20)$$

The leading contribution comes from the critical point $z_c = i(\frac{\pi}{2} - \epsilon)$ where $\epsilon$ is of order $\epsilon = o(\tilde{s}^{-1})$. We will see that in the present situation, the average $s$ is large for a macroscopic black hole, but even for moderate $s$, the leading and subleading order results would not be changed by the presence of $\epsilon$, so we will neglect it in the following. The action (19) at the critical point is

$$\mathcal{I}_\infty = \sqrt{\frac{2}{\pi}}\frac{1}{32s^3\sqrt{n}}\left(\frac{2s}{e}\right)^n \exp\left(\frac{a_H}{4} - i\frac{\pi}{2}\right), \quad (21)$$

with $a_H = 8\pi ns$ the super area of the boundary in the monochromatic case (see Equation (13)) and $e$ is Euler's constant. The analytic continuation of the state sum acquires an additional complex phase which seems counterintuitive. This is, however, in analogy to the bosonic theory [16] where it is argued that one instead needs to consider the modulus of (21). Doing so, for indistinguishable punctures the entropy $S := \ln(|\mathcal{I}_\infty|/n!)$ of the boundary is given by

$$S = \frac{a_H}{4l_p^2} + \nu \ln\left(\frac{2\sigma}{\nu}\right)\frac{\sqrt{a_H}}{l_p} - 2\ln\left(\frac{a_H}{l_p^2}\right) + \mathcal{O}(1), \quad (22)$$

where $\nu, \sigma > 0$ are numerical coefficients such that in the macroscopic limit

$$n = \nu\frac{\sqrt{a_H}}{l_p}, \qquad s = \sigma\frac{\sqrt{a_H}}{l_p}. \quad (23)$$

This scaling can be justified, and $\nu$ and $\sigma$ determined, by statistical arguments adapted from [16] which we will present in Section 4.4.

We note that the highest order term in (22) exactly reproduces the Bekenstein–Hawking area law.

No choices of parameters had to be made, unlike in the bosonic, non-chiral theory, where the Immirzi-parameter has to be adjusted to obtain the correct prefactor. This

parallels the result of [16] and hints that the chiral variables have some special properties also in the quantum theory.

### 4.3. The Multi-Color Case

Let us finally very briefly summarize the case of $j$ varying over the punctures. The analytical continuation of (17) is then given by

$$
\mathcal{I}_\infty = \frac{1}{2\pi} \int_{\mathcal{C}} dz\, \mu(z) \exp\left( \sum_{l=1}^{p} n_l \ln\left( \frac{\cos(\tilde{s}_l z)}{\cosh z} \right) \right) \times
$$
$$
\times \left[ 4 - n - \sum_{i=1}^{p} n_i \tilde{s}_i \frac{\tan(\tilde{s}_i z)}{\tanh z} \right]. \tag{24}
$$

The value of the integral in the limit of macroscopic black holes is ruled by the critical points of the action

$$
\mathcal{S}(z) := \sum_{l=1}^{p} \nu_l \ln\left( \frac{\cos(\tilde{s}_l z)}{\cosh z} \right). \tag{25}
$$

Again, we use the method of steepest decent. For more details see [14]. One finds that the entropy associated to the boundary is, at highest order, given by

$$
S = \ln(|\mathcal{I}_\infty|/n!) = \frac{a_H}{4 l_p^2} + \dots. \tag{26}
$$

Again, the entropy follows the Bekenstein–Hawking relation without the tuning of any free parameters.

### 4.4. Grand Canonical Ensemble

In [42], it has been argued that a certain class of observers near an isolated horizon would assign a quasilocal energy

$$
E = \frac{a_H}{8\pi \ell} \tag{27}
$$

where $\ell$ measures the distance of the observers to the horizon. This opens the possibility to discuss the quantum statistical mechanics of the black hole and derive its thermodynamics. If we assume that (27) is also valid for boundaries with the boundary condition (8) in supergravity, we can do the same. We will work in the grand canonical ensemble and follow the calculation in [16] closely. In particular, we will also assume indistinguishable punctures. For simplicity, we will just consider the monochromatic case. The results in the supersymmetric situation are broadly the same as for the bosonic black hole. The differences stem from the slight modifications that (22) carries as compared to the analogous formula for the bosonic black hole. It will result in changes to the coefficients (23). Starting point is the grand canonical partition function

$$
Z(\beta) = \int ds\, Z_s(\beta), \qquad Z_s(\beta) = \sum_{n=0}^{\infty} \frac{N(n,s)}{n!} e^{-\beta E(n,s)}, \qquad E(n,s) = \frac{n s l_P^2}{\ell}, \tag{28}
$$

where $N$ is the number of states. From (21), we obtain

$$
N(n,s) \approx \frac{1}{\sqrt{n s^3}} \left( \frac{2s}{e} \right)^n e^{2\pi n s} \tag{29}
$$

for large $n$. The differences compared to the bosonic case are the numerical prefactors in the exponential function and in the $n$th power. Making the definition

$$
q = \frac{2s}{e} e^{-xs}, \qquad x = \beta \frac{l_P^2}{\ell} - 2\pi, \tag{30}
$$

one can write

$$Z_s(\beta) \approx \frac{1}{s^3} \sum_{n=1}^{\infty} \frac{1}{\sqrt{n}} \frac{q^n}{n!} \tag{31}$$

as in [42].[5] Again, there is a phase transition at

$$\beta = \beta_U \equiv \frac{2\pi\ell}{l_P^2}, \tag{32}$$

at which the partition function diverges. Near the phase transition, $q$ can become very large. Indeed the maximum goes as $\max_s q \simeq 2/e^2 x$. We will analyze this regime, because arguments can be given that it coincides with the approach to the thermodynamic limit [42]. The large-$q$-expansion of the partition function gives $Z_s(\beta) \approx \exp(q)/\sqrt{q}s^3$ [42] and hence

$$Z(\beta) \approx \int ds \frac{1}{s^3} \frac{e^q}{\sqrt{q}} = \sqrt{\frac{e}{2}} \int ds \frac{1}{s^{7/2}} \exp\left[\frac{xs}{2} + \frac{2s}{e} e^{-xs}\right] \tag{33}$$

$$= \sqrt{\frac{e}{2}} x^{\frac{5}{2}} \int du (-\ln u)^{-\frac{7}{2}} u^{-\frac{3}{2}} \exp\left[-\frac{2}{ex} u \ln u\right] \tag{34}$$

$$\approx \sqrt{\frac{\pi}{2}} e^2 x^3 \exp\left(\frac{2}{e^2 x}\right) \tag{35}$$

$$= \sqrt{\frac{\pi}{2}} e^2 \frac{l_p^6}{\ell^3} (\beta - \beta_U)^3 \exp\left[\frac{2}{e^2} \frac{\ell}{l_P^2} \frac{1}{\beta - \beta_U}\right], \tag{36}$$

where the second line the substitution $s \mapsto u = \exp -xs$ was made and, in the second to last line, a saddle point approximation around the critical point $u = e^{-1}$ was made.

Now we can calculate the statistical averages $\langle a_H \rangle$, $\langle s \rangle$, and $\langle n \rangle$ of the horizon area, the color, and the number of punctures. First off

$$\langle a_H \rangle = -8\pi\ell \frac{\partial}{\partial\beta} \ln Z(\beta) = \frac{16\pi}{e} \frac{\ell^2}{l_P^2} \frac{1}{(\beta - \beta_U)^2} + \mathcal{O}((\beta - \beta_U)^{-1}). \tag{37}$$

For the mean color, we have to integrate:

$$\langle s \rangle = \frac{1}{Z_s(\beta)} \int ds\, s\, Z_s(\beta) \tag{38}$$

$$= \frac{1}{Z_s(\beta)} \sqrt{\frac{e}{2}} x^{\frac{3}{2}} \int du (-\ln u)^{-\frac{9}{2}} u^{\frac{3}{2}} \exp\left[-\frac{2}{ex} u \ln u\right] \tag{39}$$

$$\approx \frac{1}{Z_s(\beta)} \sqrt{\frac{\pi}{2}} e^2 x^2 \exp\left[\frac{2}{e^2 x}\right] \tag{40}$$

$$= \frac{1}{x} \tag{41}$$

$$= \frac{\ell}{l_P^2} \frac{1}{\beta - \beta_U}, \tag{42}$$

where again a saddle point approximation was used. Comparison with (37) gives

$$\langle s \rangle = \sigma \frac{\sqrt{\langle a_H \rangle}}{l_p}, \qquad \sigma = \frac{1}{4} \sqrt{\frac{e}{\pi}}. \tag{43}$$

Finally, for the average number of punctures

$$\langle n \rangle = \frac{1}{Z(\beta)} \int ds \sum_{n=0}^{\infty} n \frac{N(n,s)}{n!} e^{-\beta E(n,s)} \tag{44}$$

$$\approx \frac{1}{Z(\beta)} \int ds \frac{\sqrt{q}}{s^3} e^q \tag{45}$$

$$= \frac{1}{Z(\beta)} \sqrt{\frac{2}{e}} \int ds \frac{1}{s^{5/2}} \exp\left[-\frac{xs}{2} + \frac{2s}{e} e^{-xs}\right] \tag{46}$$

$$= \frac{1}{Z(\beta)} \sqrt{\frac{2}{e}} x^{\frac{3}{2}} \int du (-\ln u)^{-\frac{5}{2}} \frac{1}{\sqrt{u}} \exp\left[-\frac{2}{ex} u \ln u\right] \tag{47}$$

$$\approx \frac{1}{Z(\beta)} \sqrt{2\pi} x^2 \exp\left(\frac{2}{e^2 x}\right) \tag{48}$$

$$= \frac{2}{e^2} \frac{1}{x} \tag{49}$$

$$= \frac{2}{e^2} \frac{\ell}{l_P^2} \frac{1}{\beta - \beta_U}. \tag{50}$$

Comparison with (37) gives

$$\langle n \rangle = \nu \frac{\sqrt{\langle a_H \rangle}}{l_p}, \qquad \nu = \frac{1}{2} \frac{1}{\sqrt{\pi e^3}}. \tag{51}$$

Thus, as observed in [42] for the bosonic case, the assumption (27) allows us to rigorously justify (23). This, in turn, gives a subleading correction in the entropy $S$ that is proportional to $\sqrt{a_H}$, see (22).

## 5. Discussion and Outlook

It has been shown that in loop quantum gravity based on non-chiral real variables, apparent horizons (isolated horizons, to be precise) carry surface degrees of freedom whose entropy, in the quantum theory, is proportional to the area of the horizon [6–13]. The constant of proportionality can be set to the Bekenstein–Hawking value by a judicious choice of the free parameter of the theory, the Immirzi parameter.

However, when analytically continuing the state counting function in Immirzi parameter and state quantum numbers in a way suggesting a transition from real variables to the chiral Ashtekar variables, the correct factor in the area-entropy relation is recovered without any fine tuning [15,16].

Here we have explained that a similar result holds true in $\mathcal{N} = 1$ supergravity quantized with loop quantum gravity methods [14]. The isolated horizons are now replaced by boundaries with a boundary condition that preserves local supersymmetry on the boundary. The calculations strongly suggest that the boundary theory has an entropy $S = A/4$—a quarter of its super-area in Planck units. We also have considered the statistical physics of the boundary degrees of freedom for the first time, assuming an area-energy relation as in the non-supersymmetric theory. We were able to show that the grand canonical ensemble exists below a critical temperature and compute expectation values for the area, spins, and number punctures. These results are again analogous to those of the non-supersymmetric theory.

These results were by no means assured from the get-go. The first non-trivial result was that the boundary theory was a Chern–Simons theory precisely for the chiral case, and that the requirement of supersymmetry on the boundary uniquely fixes boundary conditions (8) that are structurally similar to those of the non-supersymmetric theory. The next important puzzle piece is the existence of a compact real form of $OSp(1|2)_{\mathbb{C}}$. We were able to calculate a Verlinde-type formula explicitly for the case of a large level. Moreover, there exist $OSp(1|2)_{\mathbb{C}}$ representations with the real area, and the corresponding

analytic extension of the formula for the number of states makes sense and yields the Bekenstein–Hawking relation.

We note that Chern–Simons level is proportional to the inverse of the cosmological constant. Therefore the large-$k$ limit makes physical sense; we could deal with the classical superalgebra instead of a quantum supergroup. We also note that there is a slight change in the leading order term in the number of states, as compared to the non-supersymmetric theory: there appears an additional factor of two in the exponent, but it can be interpreted simply as the area spectrum for two-sided punctures.

A further difference to the bosonic case is that the boundary conditions are not directly tied to the boundary being an isolated horizon. One can, however, speculate that the boundary condition in the bosonic case is only a necessary condition, not sufficient for the existence of an isolated horizon (see, for instance, the discussion in [43,44]) and that the boundary condition we considered can be shown to be necessary for the existence of some sort of apparent horizon through future work.

There is a further issue with our calculation that needs greater understanding. The coupling of the bulk excitations to the boundary theory can be interpreted as required gauge invariance, which in turn is equivalent to the implementation of the left supersymmetry constraint. The right-handed supersymmetry constraint was not implemented directly. However, one can speculate that it does not reduce the number of surface states in the highest order. This is very similar to the assumption for the Hamilton constraint in the bosonic theory, which is also not implemented. In addition, we have only implemented the reality conditions in the sense of analytically continuing to representations with real areas. It is possible that more is necessary. These issues warrant further attention.

Additionally, the results should be generalized to include extended supersymmetry, $\mathcal{N} > 1$. Ref. [19] provides a good understanding of how the bulk theory would look like for $\mathcal{N} = 2$. This could be a basis to extend the entropy calculation to more interesting theories with $\mathcal{N} = 2$. Complementary to this, one would like to understand how to describe the horizons of BPS black holes and their boundary theory in loop quantum gravity and attempt a calculation of their entropy to compare with the treatment in string theory [4,5]. Additionally, $OSp(m|n)_{\mathbb{C}}$ super Chern–Simons theories with a complex level $k$ are found as boundary theories in string theory [41]. We would like to understand the possible connections of [41] to the present work in general, and in particular between the analytic continuation considered there and the one we use.

**Author Contributions:** Conceptualization, K.E. and H.S.; formal analysis, K.E. and H.S.; methodology, K.E. and H.S.; Writing—original draft, except Sections 1 and 4.4, K.E.; Writing—original draft Sections 1 and 4.4, H.S.; Writing—review and editing, K.E. and H.S. All authors have read and agreed to the published version of the manuscript.

**Funding:** This research received no external funding.

**Data Availability Statement:** Data sharing not applicable.

**Acknowledgments:** We would like to thank Lee Smolin for communications at an early stage of this work and in particular for his interest in application of loop quantum gravity methods to supersymmetric black holes which was part of the motivation of this work. We thank the anonymous referees for their careful reading of the manuscript and their helpful suggestions to improve the presentation. K.E. thanks the German Academic Scholarship Foundation (Studienstiftung des Deutschen Volkes) for financial support. H.S. acknowledges the contribution of the COST Action CA18108.

**Conflicts of Interest:** The authors declare no conflict of interest.

## Notes

[1] The area with respect to the bosonic component of the super-geometry is not gauge invariant and, therefore, not observable. However, in a gauge in which the superelectric field has vanishing odd components, the super area, and the bosonic area would agree.

[2] This is actually the case for both, $\mathcal{N} = 1$ and 2, [19], but here we consider $\mathcal{N} = 1$ only.

[3]　Here, a capital letter is used to denote the Majorana fermion field (containing both chiralities). The chiral subcomponents are denoted, respectively, by lower case letters $\psi_r$ and $\psi^r$ (with the position of the $R$-symmetry index explicitly indicating the chirality).

[4]　We remind the reader that $\Delta$ is the boundary of the spatial slice $\Sigma$, i.e., a spatial slice of the boundary $H$ of spacetime manifold $M$.

[5]　Here we have started summation at $n = 1$, since the approximation (29) is ill-defined for $n = 0$. It is valid only for large $n$ anyway, and the low $n$ contributions do not make a difference in the macroscopic regime, but it is convenient to keep them to obtain closed-form expressions in the following.

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
