# Peer review of "Chiral Loop Quantum Supergravity and Black Hole Entropy"

_universe, doi:10.3390/universe9070303_

Round 1

Author Response

Thanks a lot for your careful reading of the text and for the helpful suggestions. 

We have removed inconsistencies in the numbering of the references.  
The parameter $e$ is Euler's constant. We have put a remark in the text.  
Indeed the expectation value brackets were missing in (43) and also in (51), we have corrected that.

We have made some other changes, a list of changes (except for removal of typos) is below.

We hope you judge the manuscript to be ready for publication now. 

List of changes

- Literature reordered
- dots and commas after equations consistently placed
- added "and $e$ is Euler's constant" after (21)
- added expectation value brackets in (43)
- added expectation value brackets in (51)
- summation variable in (31) changed to n
- changed $N$ to $n$ in (51)

- added explanation of symbols to text around (2) 
- added explanation of all the ingredients of (3) below (3)
- rendered $N$ blackboard bold below (11)
- introduced manifold $M$, its boundary $H$ above (1), defined $\Delta$ above (2).
- added footnote 4 to remind reader of definition of $\Delta$  
- added explanation of $\alpha$ below (15). 
- stated definition of $n$ above (17)
- cited [14] before (17) and (18).
- text below (18) and (20) modified to better justify the macroscopic limit and the roles played by $n$ and $s$ in the approximation. 

- made clearer that (29) is an approximation, added footnote 5 to explain change in summation. 
- added a word in first sentence of 3rd par of sec 5.

- added footnote 3 with an explanation of different notations used for Majorana and chiral fermions
- explained below (18) how the relevant term for the saddle point approximation is determined. 
- added references in paragraph 6 of section 5. 

Reviewer 2 Report

In this article, the authors review the calculation of the entropy associated to a boundary in the context of the supersymmetric version of loop quantum gravity with N = 1. The ultimate result is the derivation of the Bekenstein-Hawking area law at leading order without any parameter fine-tuning, provided that one imposes a specific boundary condition induced from the requirement of local supersymmetry at the boundary. The methods used and the results obtained are very interesting, and they provide new avenues to investigate the black hole entropy calculations in quantum gravity and to explore the connections between string theory and supersymmetric loop quantum gravity. I therefore recommend the article for publication, pending the following amendments:

1) It would be helpful for the readers if the notation used in some equations is explained. Namely:
- equation (2): mention that F is the curvature of the connection, E is the super-electric field (which could be added in the paragraph preceding the equation where it is first mentioned) and the arrows denote the pullback to the boundary.
- equation (3): mention what every element is on the right-hand side.
- in the text below equation (11), the set of natural numbers needs to be denoted properly as it is done later in the text.
- equation (15): mention what are ∆ (the 2 dimensional boundary?) and α (arbitrary smearing function?).
- in the text above equation (17), specify what is n (I am assuming it is the total number of punctures).

2) Add the reference [1] above equations (17) and (18) for the details of the calculations.

3) Could the authors include, in the text below equation (18), the justification for defining the limit of macroscopic black holes as  s → ∞ and n → ∞?

4) In equation (28), shouldn't the sum start from n=1 as in equation (31), since N(n,s) in (29) diverges for n=0? Also, in equation (31), the label in the sum should be n, not k.

5) Just for clarity, please add a space between ds and s in equation (38).

6) In equations (43) and (51), a_H should be replaced by the average <a_H>.

7) The word "result" is missing in the first sentence of the third paragraph of section 5: "... a similar result holds true ..."

There are few typos and missing (grammatical) articles throughout the text, please revise that.

Author Response

Thanks a lot for your careful reading of the text and for the helpful suggestions. 

Regarding your point 1), we added the explanations that were missing. 

Regarding your point 2), we added the reference. 

Regarding your point 3), indeed the text was not very clear. We have now explained that and n → ∞ is natural for macroscopic black holes, and is the one that makes the stationary state approximation applicable, whereas the limit of large s is used to give a closed form expression for the most relevant critical point. We also point out that the terms that we drop limit of large s would not change the leading order and next to leading order terms in our calculations. 

Regarding your point 4) again, the text was not very clear. (28) is an exact expression for the grand canonical partition function. The problem with $n=0$ only appears when $N$ is  replaced by an approximation valid for large n. We have explained the situation in footnote 5.    

Regarding your points 5),6),7), we have made the requested changes.  

We have made some other changes, a list of changes (except for removal of typos) is below.

We hope you judge the manuscript to be ready for publication now. 

List of changes

- Literature reordered
- dots and commas after equations consistently placed
- added "and $e$ is Euler's constant" after (21)
- added expectation value brackets in (43)
- added expectation value brackets in (51)
- summation variable in (31) changed to n
- changed $N$ to $n$ in (51)

- added explanation of symbols to text around (2) 
- added explanation of all the ingredients of (3) below (3)
- rendered $N$ blackboard bold below (11)
- introduced manifold $M$, its boundary $H$ above (1), defined $\Delta$ above (2).
- added footnote 4 to remind reader of definition of $\Delta$  
- added explanation of $\alpha$ below (15). 
- stated definition of $n$ above (17)
- cited [14] before (17) and (18).
- text below (18) and (20) modified to better justify the macroscopic limit and the roles played by $n$ and $s$ in the approximation. 

- made clearer that (29) is an approximation, added footnote 5 to explain change in summation. 
- added a word in first sentence of 3rd par of sec 5.

- added footnote 3 with an explanation of different notations used for Majorana and chiral fermions
- explained below (18) how the relevant term for the saddle point approximation is determined. 
- changed text in last paragraph of sec 4.2
- $u^{3/2}$ replaced by $u^{-3/2}$.
- added references in paragraph 6 of section 5. 
- changed last three sentences of abstract.

- acknowledgement of the referees

Reviewer 3 Report

The paper presents a review of the calculation of the entropy for the boundary in the supersymmetric version of loop quantum gravity for the minimal case N=1, using the super Chern-Simons theory. The authors calculate the dimensions of the quantum state spaces of UOSp(1|2) super Chern-Simons theory with punctures and its analytical continuation to OSp(1|2)C, for fixed quantum super area of the surface, resulting in S=aH/4 for large (super) areas and determining lower order corrections. They also initiate a discussion on the statistical mechanics of the surface degrees of freedom.

The paper is well-written, and the calculations are sound. The authors provide a thorough discussion of their results and their significance. The proposal to define and calculate the entropy for the boundary in the supersymmetric version of loop quantum gravity for the minimal case N=1 via the super Chern-Simons theory is interesting and significant for the field.

Overall, the paper makes a valuable contribution to the field of supersymmetric loop quantum gravity and provides new insights into the calculation of entropy for the boundary. As such, I recommend the acceptance of this manuscript for publication in its current form.

Author Response

Thanks a lot for your careful reading of the text and for the positive evaluation.

We have made some changes in response to other referees. A list of changes (except for removal of typos) is below for your information.  

List of changes

- Literature reordered
- dots and commas after equations consistently placed
- added "and $e$ is Euler's constant" after (21)
- added expectation value brackets in (43)
- added expectation value brackets in (51)
- summation variable in (31) changed to n
- changed $N$ to $n$ in (51)

- added explanation of symbols to text around (2) 
- added explanation of all the ingredients of (3) below (3)
- rendered $N$ blackboard bold below (11)
- introduced manifold $M$, its boundary $H$ above (1), defined $\Delta$ above (2).
- added footnote 4 to remind reader of definition of $\Delta$  
- added explanation of $\alpha$ below (15). 
- stated definition of $n$ above (17)
- cited [14] before (17) and (18).
- text below (18) and (20) modified to better justify the macroscopic limit and the roles played by $n$ and $s$ in the approximation. 

- made clearer that (29) is an approximation, added footnote 5 to explain change in summation. 
- added a word in first sentence of 3rd par of sec 5.

- added footnote 3 with an explanation of different notations used for Majorana and chiral fermions
- explained below (18) how the relevant term for the saddle point approximation is determined. 
- changed text in last paragraph of sec 4.2
- $u^{3/2}$ replaced by $u^{-3/2}$.
- added references in paragraph 6 of section 5. 
- changed last three sentences of abstract. 

- acknowledgement of the referees

Reviewer 4 Report

This work summarizes recent results about the loop quantization of $N$=1 supergravity using chiral Ashtekar variables and about the entropy of the boundary that plays the role of isolated horizon. The degrees of freedom on this boundary are described by a super Chern-Simons theory. A condition preserving supersymmetry on this boundary relates those degrees of freedom with the bulk ones. Although the relevant Chern-Simons theory has an imaginary level and a noncompact structure group, the dimension of the quantum space of states is derived by analytical continuation, leading to an entropy that satisfies the Bekenstein-Hawking area law. In addition, the article studies the statistical physics of the boundary degrees of freedom, constructing a grand canonical ensemble and computing with it expectation values for several quantities.

The article is well written and gives a comprehensible summary of the most important results on this topic. I have several comments that the authors must take into account before I can recommend their work for publication.

An important point is that they should provide a more detailed argument supporting that (19) is the relevant action for the steepest-descent evaluation of the partition function, and that there is no other exponential contribution to the integrand that needs to be analyzed. Besides, the authors should make clear which results are new compared to previous work, stating this in the abstract and the introduction. Concerning also the abstract, the present paper may be an overview rather than a review of previous work. On the other hand, at the end of Sec. 4.2, the fact that the Immirzi parameter does not appear in the calculation is not so surprising, because $\beta$ has already been taken equal to $-i$ in the construction and Eq. (11) shows that the area eigenvalues for $a_H$ are independent of that parameter. In Sec. 4.3, the paragraph after Eq. (25) does not contain much information and only repeats results from Ref. [1], therefore it can be shortened. In addition, at the end of the sixth paragraph of Sec. 5, it would be helpful that the reference to possible future work could be a bit more specific.

Apart from this, in the limit $s\rightarrow\infty$ and $n\rightarrow\infty$ introduced below Eq. (18), the authors should explain that the ratio of these two quantities must be restricted to a constant and argue why. Besides, they should check the notation used for the last terms of Eq. (3) and Eq. (5), and clarify the relation between these terms if necessary. The symbol $\alpha$ in Eq. (15) and the notation $\nu_l$ in Eq. (25) need to be explained. In Eq. (31), $k$ should read $n$. In Eq. (34), the power $3/2$ most probably is $-3/2$. In Eq. (37), the last term is of order $-3$ and not $-1$ (as a power of $\beta-\beta_U$). In Eqs. (38), (39) and (40), the symbol $Z_s(\beta)$ in the denominators should probably be $Z(\beta)$. In Eq. (39), the power of $(-\ln{u})$ seems to be $-5/2$, and the power of $u$ is $-3/2$. In Eq. (51), the first expectation value is $<n>$, not $<N>$. And the denominators of several mathematical expressions in the text should be in parentheses: page 3, line 101; page 6, line 205; and page 8, last line.

Finally, references should be listed in the order in which they appear in the text. 

There are some misprints that the authors should correct: ``On the other hand, but many aspects’’ (page 2, line 57);  an extra parenthesis after ``constructed in [1]’’ (page 4, line 134); the wording “disk” (page 5, line 177);  ``a suitable functions’’ (page 5, line 179); ``these spanning’’ (page 6, line 214); `` The results in a subleading correction in the entropy’’ (page 10, line 285); possibly `` explained that a similar holds true’’ (page 10, line 297); `` sot sufficient’’ (page 11, line 324); and `` assumptionfor’’ (page 11, line 333). 

Author Response

Thanks a lot for your careful reading of the text and for the helpful suggestions. 

We will list your comments and our response: 

1) "should provide a more detailed argument supporting that (19) is the relevant action for the steepest-descent evaluation of the partition function, and that there is no other exponential contribution to the integrand that needs to be analyzed." 

We have given the argument below (18)

2) "the authors should make clear which results are new compared to previous work, stating this in the abstract and the introduction. Concerning also the abstract, the present paper may be an overview rather than a review of previous work."

We have made changes in the abstract accordingly. 

3) "at the end of Sec. 4.2, the fact that the Immirzi parameter does not appear in the calculation is not so surprising, because $\beta$ has already been taken equal to $-i$ in the construction and Eq. (11) shows that the area eigenvalues for $a_H$ are independent of that parameter.

Good point, We have changed the text in the last paragraph of sec 4.2 accordingly. 

4) "In Sec. 4.3, the paragraph after Eq. (25) does not contain much information and only repeats results from Ref. [1], therefore it can be shortened."

We are not sure what is meant here. The paragraph is short and should not be shortened further. Perhaps a different paragraph was meant?

5) "In addition, at the end of the sixth paragraph of Sec. 5, it would be helpful that the reference to possible future work could be a bit more specific."

We have added references to a more detailed discussion. 

6) "Apart from this, in the limit $s\rightarrow\infty$ and $n\rightarrow\infty$ introduced below Eq. (18), the authors should explain that the ratio of these two quantities must be restricted to a constant and argue why."

Indeed the description of the macroscopic limit was not very clear. We have rewritten the text below (18), we think this takes care of the request. 

7) "Besides, they should check the notation used for the last terms of Eq. (3) and Eq. (5), and clarify the relation between these terms if necessary."

We have explained the notation in a new footnote, footnote 3. 

8) "In Eq. (34), the power $3/2$ most probably is $-3/2$."

Yes, thanks for catching that. we corrected it. 

9) "In Eq. (37), the last term is of order $-3$ and not $-1$ (as a power of $\beta-\beta_U$)."

We think that (37) is correct as it stands: $\beta-\beta_U$ is small and $<a_H>$ is diverging for $\beta=\beta_U$, so the leading order term (from the exponential in (36)) is order $-2$ and the next order (from the $(\beta-\beta_U)^3$ in (36)) is order $-1$.  

10) "In Eq. (51), the first expectation value is $<n>$, not $<N>$."

Right. fixed. 

11) "the denominators of several mathematical expressions in the text should be in parentheses: page 3, line 101; page 6, line 205; and page 8, last line." 

We think that it is generally understood that in inline equations, everything after "/" and until the next " " is in the denominator. Therefore we left the equations as they are for now. If you or Universe editorial staff insists, we will change it. 

11) "Finally, references should be listed in the order in which they appear in the text." 

We fixed the inconsistencies in the numbering.  

Also thank you for the "Comments on the Quality of English Language". We have made the corrections as you requested. 

We have made some other changes, a list of changes (except for removal of typos) is below.

We hope you judge the manuscript to be ready for publication now. 

List of changes

- Literature reordered
- dots and commas after equations consistently placed
- added "and $e$ is Euler's constant" after (21)
- added expectation value brackets in (43)
- added expectation value brackets in (51)
- summation variable in (31) changed to n
- changed $N$ to $n$ in (51)

- added explanation of symbols to text around (2) 
- added explanation of all the ingredients of (3) below (3)
- rendered $N$ blackboard bold below (11)
- introduced manifold $M$, its boundary $H$ above (1), defined $\Delta$ above (2).
- added footnote 4 to remind reader of definition of $\Delta$  
- added explanation of $\alpha$ below (15). 
- stated definition of $n$ above (17)
- cited [14] before (17) and (18).
- text below (18) and (20) modified to better justify the macroscopic limit and the roles played by $n$ and $s$ in the approximation. 

- made clearer that (29) is an approximation, added footnote 5 to explain change in summation. 
- added a word in first sentence of 3rd par of sec 5.

- added footnote 3 with an explanation of different notations used for Majorana and chiral fermions
- explained below (18) how the relevant term for the saddle point approximation is determined. 
- changed text in last paragraph of sec 4.2
- $u^{3/2}$ replaced by $u^{-3/2}$.
- added references in paragraph 6 of section 5. 
- changed last three sentences of abstract. 

- acknowledgment of the referees

Round 2

Reviewer 4 Report

The manuscript has certainly been improved with the modifications introduced by the authors. I consider that it can be accepted for publication in its present form. I agree with the authors that no correction is nedded concerning point (9) of my previous report. The authors may want to consider the following minor points. In lines 47 and footnote 4, it would be more accurate to replace ``spacetime’’ with ``spacetime manifold’’. In line 89, ``the $T$ generate the R-symmetry,’’ could be corrected to ``and $T$ generate the R-symmetry.’’ (in particular, note the final period). In line 306, ``The results in a subleading correction’’ probably means ``The results give a subleading correction’’. Finally, in line 352, ``the highest order’’ could be changed to ``the leading order’’.  

The English is correct. Minor editing required.

Author Response

Thank you for the careful reading and for the suggestions. We have implemented them all.